# Post-Intervention Reconstruction and the Responsibility to Rebuild

**Athanasios Stathopoulos**

International Studies, Leiden University, 2511 VA The Hague, The Netherlands;
a.stathopoulos@hum.leidenuniv.nl

**Abstract:** This article examines the relationship between the responsibility to rebuild and post-intervention reconstruction. It aims to determine whether the current interpretation of the responsibility to rebuild is the appropriate framework for attaining the goals of post-intervention reconstruction. The article argues that, despite the urgent need for a post-intervention strategy in the aftermath of humanitarian interventions, the responsibility to rebuild, as it is currently being framed, can end up undermining the goals of post-conflict reconstruction by dissuading states from participating in atrocity prevention, inadvertently increasing atrocity crimes and delegitimizing military humanitarian interventions. The analysis identifies the need for the responsibility to rebuild to incorporate an increased respect for *post bellum* proportionality and self-determination.

**Keywords:** responsibility to protect; post-conflict reconstruction; *jus post bellum*

## 1. Introduction

In 2001, the International Commission on Intervention and State Sovereignty (ICISS)[1], published its report entitled *The Responsibility to Protect*. The commission had been tasked with tackling the issue of intervention for human protection purposes, "one of the most controversial and difficult of all international relations questions" as it noted (ICISS 2001, p. vii). The report concluded that states have a responsibility to protect their citizens, with that protection including a responsibility to prevent, to react, and to rebuild. In 2005, when the responsibility to protect (RtoP) was unanimously adopted by the UN Member States at the World Summit, this responsibility was specified to apply to genocide, war crimes, ethnic cleansing, and crimes against humanity. Ever since, the literature on the responsibility to protect has proliferated, focusing on the evolution (Evans 2015) and adoption (Bellamy 2008) of the norm, its link with other norms (Rhoads and Welsh 2019), and its preventive potential (Gallagher 2015), among a plethora of works.

The responsibility to rebuild was one of the three main elements of states' responsibility to protect in the 2001 ICISS report and the report specified that the "responsibility to protect implies the responsibility not just to prevent and react, but to follow through and rebuild. This means that if military intervention action is taken—because of a breakdown or abdication of a state's own capacity and authority in discharging its "responsibility to protect"—there should be a genuine commitment to helping to build a durable peace, and promoting good governance and sustainable development" (ICISS 2001, p. 39). In the seven pages that the report devotes to states' post-intervention obligations, the various priorities are outlined, ranging from the reconstitution of public safety and order by international agents in cooperation with local people and the subsequent transfer of authority, to the investment of adequate time and resources, and various pitfalls that need to be avoided such as lack of local ownership, dependency, and undermining of sovereignty (ICISS 2001, pp. 39–45).

Despite the importance that the ICISS commissioners placed on the responsibility to rebuild, this element has been subsequently largely ignored both by policymakers and

scholars (Keranen 2016, p. 333). The responsibility to rebuild was not included in the paragraphs of the 2005 World Summit Outcome Document. Instead, peacebuilding was decoupled from RtoP and, at the 2005 World Summit, the Peacebuilding Commission was established with the aim to "propose integrated strategies for post-conflict[2] peacebuilding and recovery" [ . . . ] and "focus attention on the reconstruction and institution-building efforts necessary for recovery from conflict" (United Nations 2005). The establishment of the Peacebuilding Commission in principle can be considered an important achievement of the 2005 World Summit; however, the decoupling of the two agendas "despite the strong normative and function overlap between the two" (Bellamy 2021, p. 278) can also be seen as undermining the overall goals of the responsibility to protect.

Until recently, the question of what needs to be done and the policies that should be followed after an intervention has taken place have to a great extent been ignored. Most of the discussion has focused on the details for the implementation of the responsibility to rebuild (e.g., who should do the rebuilding (Pattison 2013) and the rationale behind state rebuilding narratives (Gheciu and Welsh 2009)), and has eschewed a critical examination of whether the constitutive elements and current priorities of the responsibility to rebuild can have a positive contribution to the goals of a military humanitarian intervention more broadly, and post-conflict reconstruction in particular, and what its possible negative effects might be[3]. This article aims to fill this void and to answer the question of whether the responsibility to rebuild—as it is currently being interpreted—is the appropriate framework for post-intervention reconstruction. It will be shown that the responsibility to rebuild, as it was framed in the 2001 ICISS report and as it is currently being interpreted by the majority of scholars, despite its good intentions in practice can undermine the goal of post-conflict reconstruction by dissuading states from participating in atrocity prevention and inadvertently increasing atrocity crimes, delegitimizing military humanitarian interventions, and inflicting disproportionate harm in the military humanitarian interventions conducted.

## 2. Unpacking the Responsibility to Rebuild

Before delving into the various elements of the responsibility to rebuild and the possible effects of its implementation, it is important to underline the importance of post-conflict reconstruction and its relationship with the goals of humanitarian intervention. The responsibility to rebuild should be seen as an indispensable component of humanitarian interventions and a sine qua non for the establishment of a stable and durable peace. The ICISS report rightly pointed out that in the past "the exit of the interveners has been poorly managed, the commitment to help with reconstruction has been inadequate, and countries have found themselves at the end of the day still wrestling with the underlying problems that produced the original intervention action" (ICISS 2001, p. 39). The report was correct in underlying the need for a post-intervention strategy, with the objectives of that strategy being to eliminate the conditions that made the military intervention necessary in the first place. Despite the recent interest in *jus post bellum*, states' *post bellum* responsibilities are not a recent development, with Augustine arguing that war can only be justified if it is fought to achieve a just peace and combatants being obliged to account not just for their respect of *ad bellum* and *in bello* responsibilities, but the way that the war ends and the transition to peace is made (Augustine 1972, p. 866; Bellamy 2008, p. 602). Although post-conflict obligations have often been ignored, it has become clear that achieving the goals of a military intervention cannot be done without a post-intervention strategy; thus, the ICISS report was correct in underlying the need for states to invest in their post-intervention obligations.

Yet, despite the overarching consensus on the necessity for a post-intervention strategy, the rationale and specific methods of that strategy are often in contention. Gheciu and Welsh (2009) have identified four imperatives behind states' rationales for rebuilding; that is, special responsibilities incurred in using force, projection of norms and values, national interest and defense of society, and restoration of self-determination. The ways through which interventions are conducted have been at the center of attention for decades and

their focus has significantly shifted. From the overly prescriptive and "one size fits all" international programs of the 1990s (Chandler 2017), the mid-2000s saw a shift towards a bottom-up approach. The top-down interventions of the 1990s, with their emphasis on sustainable development, peace, and post-conflict reconstruction through the transfer—and often imposition—of liberal institutional frameworks proved ineffective. As Roland Paris noted, the export of liberal institutions cannot easily and quickly bring results and that there "is no easy, quick, and cheap method of establishing stable and lasting peace" (Paris 2004, p. 206). Thus, in the mid-2000s a focus on national ownership and "homegrown" policies was witnessed. In the 2005 Paris Declaration on Aid Effectiveness, nationally led development strategies were prioritized. Moreover, the New Deal on Peacebuilding conceived by the International Dialogue on Peacebuilding and Statebuilding and endorsed in 2011 during the fourth High Level Forum on Aid Effectiveness is similarly underlining the importance of local ownership. It is thus important to review the focus of the responsibility to rebuild under this light.

The focus of the ICISS report's responsibility to rebuild reflects the changing priorities in post-intervention obligations in the beginning to mid-2000s. On the one hand, the report acknowledges the need to act "in partnership with local authorities" and for there to be "close cooperation with local people", and recognizes the importance of local ownership and the risks arising from the suspension of sovereignty and the creation of dependency (ICISS 2001, pp. 39–45). On the other hand, the report's section of the responsibility to rebuild shows signs of the top-down, expansive, and prescriptive programs of the 1990s. Thus, states' responsibility to rebuild includes a commitment to "helping build a durable peace, and promoting good governance and sustainable development" as well as a reconstitution of public safety and order "by international agents", adding that this should be done "in partnership with local authorities" (ICISS 2001, p. 39). Finally, in an apparent attempt to accommodate fears and accusations of neocolonialism, the report notes that the goal should be the gradual transfer of authority and responsibility to rebuild back to local authorities.

The main points of contention relate, on the one hand, to the wide-ranging nature of post-intervention obligations (durable peace, good governance, and sustainable development), which inevitably lead to overly ambitious, long, and intrusive interventions, or as Rodin notes "encompassing a permission to engage in deep intervention in the political processes of the target community" (Rodin 2014, p. 247), and on the other hand, to the possible neglect of local ownership and self-determination by an emphasis on externally designed and led programs. Regarding the former, critics underline the risk of violation of the rights and liberties of the target communities, arguing often that elements of these interventions and post-conflict reconstructions amount to neocolonialism (Wilde 2008; Caplan 2005). The fear is that such all-encompassing goals such as durable peace, good governance, and sustainable development, worthy as they may be, could be used as an excuse for protracted and infinite interventions serving the interests of intervening states. Regarding the mention of local ownership, it can be argued that the report pays lip service to the importance of nationally led strategies, as it seems to share the "assumptions of external knowledge and direction" prevalent in the 1990s, with these interventions unable to escape "new forms of exclusion and marginalization" (Chandler 2021, pp. 431–33).

The wide-ranging nature of post-intervention obligations, together with the prioritization of internationally designed and externally directed programs, can undermine the goals of post-conflict reconstruction. The next section will explore the specific mechanisms through which this may take place.

### 2.1. Undermining Post-Conflict Reconstruction

The necessity of post-conflict reconstruction on the eve of humanitarian interventions is widely acknowledged in order to address the conditions that led to the intervention and to ensure that the likelihood of atrocity crimes recurring is minimized. However, the way that post-conflict reconstruction is to be conducted is a matter of contention. The

following section will explore the link between the responsibility to rebuild and post-conflict reconstruction and will show that despite its noble intentions the responsibility to rebuild, were it to be implemented in the way it is currently being interpreted, is likely to undermine the goals of post-conflict reconstruction.

*2.2. Dissuading Intervention, Expanding Atrocities, and Increasing Harm*

2.2.1. Dissuading Intervention

According to the ICISS report, the goals under the responsibility to rebuild include durable peace, good governance, and sustainable development (ICISS 2001, p. 39). Although these are noble goals that would in theory be beneficial for RtoP as a norm, in practice such a broad interpretation of the responsibility to rebuild could be problematic. As the report itself identified, the high cost of rebuilding for participating countries could dissuade a lot of them from participating in RtoP campaigns in the future. Scholars have followed suit and adopted an equally expansive and externally determined list of obligations. Ramesh Thakur has suggested that in the aftermath of an intervention the goal should be "the restoration and development of effective, efficient and legitimate public institutions capable of delivering the necessary goods and services, assuring public safety and order, exercising monopoly control over the legitimate use of force, mediating intergroup competition, and promoting economic development" (Thakur 2018, pp. 6–7). The responsibility to rebuild, according to Hilpold, involves the vanquished state being helped to "become a peace-loving, politically, economically and legally stable member of the international society" (Hilpold 2020). On the other hand, scholarship has largely correctly identified the potential risks of additional responsibilities on states. Keohane has underlined the risk of protracted interventions with no end in sight, arguing that "cautious policy-makers will be reluctant to intervene in the first place, even when threats to human rights are severe and, on moral grounds, intervention should be threatened or used" (Keohane 2003, p. 281). Bellamy identifies a number of inhibitors of armed intervention, among them international law, prudential considerations, and political will (Bellamy 2010, p. 611). Regarding the latter, the traditional understanding is that political leaders may lack the political will to commit to a military humanitarian intervention, due to cost-benefit analysis and prioritization of domestic considerations. Bellamy, however, underlines the importance of prevailing interests, meaning that the pursuit of the state's own interests "leads them to support or shield the perpetrators" (Bellamy 2010, p. 611). Despite popular perceptions, military humanitarian interventions to end atrocity crimes are very rare, or as Bellamy puts it "[i]nternational society's default response to genocide is to stand aside and hope that the blood-letting comes to an end incidentally" (Bellamy 2010, p. 615). Robert Pape finally notes that the creation of limitless obligations in the form of nation-building, rebuilding, and establishment of "new political, economic, and social institutions" can lead to opposition to any moral action (Pape 2012, pp. 49–52).

The Rwandan genocide in 1994 constitutes an apt example of both the international society's default reaction to instances requiring a military humanitarian intervention and the effect that potentially high costs for intervening states, especially in the absence of a combination of "material self-interests and humanitarian norms" (Paris 2014, p. 573), can have on the probability of state intervention. Beginning in April 1994, and for a period of three-and-a-half months, approximately 800,000 members of the Tutsi ethnic group were killed together with some moderate Hutu. The international community largely remained on the sidelines during the genocide and failed to prevent or stop the atrocities from taking place. Initially, it was argued that the failure to respond to the Rwandan genocide was due to a lack of knowledge of the extent and nature of the brutalities. As President Clinton argued on 25 March 1998, I did not "fully appreciate the depth and the speed with which [Rwandans] were being engulfed by this unimaginable terror". However, evidence quickly mounted about the warning signs that had been given regarding the imminent catastrophe and it is now clear that "U.S. officials shunned the term 'genocide', for fear of being obliged to act" (Power 2001). The Rwandan case provides evidence of the fact that, on the one

hand, even the gravest violation of human rights can be disregarded by the international community unless the right mix of self-interest, international conditions, and humanitarian urgency is present, and most importantly, on the other hand, that the existence of positive obligations can not only be proven insufficient to nudge a state towards taking action but can also dissuade intervention if the obligations are deemed by potential interveners to be onerous and beyond the realm of politically feasible at that moment in time.

Whether states' reservations to intervene are due to cost-benefit analysis, prioritization of domestic considerations, or lack of political will, given states' reluctance towards military humanitarian interventions, the fear is that broad additional responsibilities in the form of the responsibility to rebuild would constitute a heavy burden on would-be interveners, which could end up dissuading them from participating in the intervention in the first place.

### 2.2.2. Expanding Atrocities

It is nowadays widely recognized that RtoP has become an international norm[4], shaping expectations about how states should respond to atrocity crimes[5]. The focus is not so much on whether an intervention should take place, but on how; or, as the then UN Secretary General Ban Ki-moon noted in 2011, "our debates are now about how, not whether, to implement the responsibility to protect", adding that "[n]o government questions the principle" (United Nations 2011). Even countries initially skeptical of RtoP, such as China, have endorsed it and have not hesitated to agree to the use of force as a last resort. It is thus safe to say that the notion of states having a responsibility to protect their population from atrocity crimes, as well as that the international community has a responsibility to adopt measures intended to prevent and/or respond to atrocity crimes is uncontroversial. However, this consensus can easily be overturned and its implementation should not be taken for granted. The incidence of atrocity crimes, as Bellamy and Luck underline, is again rising after more than a decade of decline, with the "tide of forcibly displaced populations [ . . . ] at its highest level since the end of World War II" (Bellamy and Luck 2018, p. 1). Dissuading countries from participating in atrocity prevention and reaction could thus have a corollary negative ripple effect, that of expanding atrocity crimes and increasing the chances of atrocity crimes being committed. This could be the result of diminished expectations that the international community would engage when faced with situations of atrocity crimes, and states either failing to use the tools for monitoring, assessing, and forecasting atrocity crimes or opting to stand aside (or being actively involved in) when atrocity crimes are being committed. Apart from states, it has been noted that nonstate actors are increasingly actively involved in atrocity crimes, with atrocity tactics reported being used "by terrorist organizations that seek to elevate their public profile through widespread and systematic attacks against civilians" (Policy Dialogue Brief 2015, p. 5).

An accusation often leveled against RtoP is that of moral hazard, the idea that responsibility to protect causes genocidal violence that would not otherwise occur (Kuperman 2009, p. 282; Belloni 2006, p. 327). According to the moral hazard theory, the promise of third-party intervention can cause groups to adopt risky behavior and protract conflicts and misery by potentially discouraging groups from negotiating a settlement. Although the application of this theory on conflict management in general and on atrocity crimes in particular appears, at first, to have merit, recent analysis undermines its validity, showing that it "provides misleading accounts of actors' motivations and intentions" and that it "is unsuited to the explanation of complex social phenomena" (Bellamy and Williams 2011, pp. 557–58). However, it can be argued that moral hazard could operate in the exact opposite way than its proponents argue. In the event of countries being dissuaded from participating due to increased risks and costs and thus leading to diminished expectations of intervention, states and/or non-state actors emboldened by the knowledge that an intervention will be unlikely may consider the use of all available tactics, including atrocity crimes, in order to achieve their political goals and to prevail over their rivals. There is a large number of examples where a regime "successfully" used atrocities to achieve their political goals and where inaction by the international community led to these practices be-

ing repeated in the future. In Syria, Hafez al-Assad's regime engaged in systematic torture, arbitrary killings, and use of artillery and airpower against the Muslim Brotherhood in the 1980s, with Bashar al-Assad's regime repeating the same strategy after 2011. Vladimir Putin's use of indiscriminate violence in Syria had also been witnessed in Chechnya. In Sudan, Omar al-Bashir's use of indiscriminate force in Darfur was repeated later on in South Kordofan (Bellamy 2018, pp. 332–33). The evidence shows that the unchecked use of atrocities may expand their occurrence as the regime is both emboldened and habituated in the use of atrocities. Thus, while the moral hazard's predictive value towards atrocity crimes is low, as the incidence of the latter is due to a multitude of factors including social and historical context, economic reasons, and local dynamics (Collier and Hoeffler 2004; Autesserre 2010; Kalyvas 2006; Bellamy and Williams 2011), it can be argued that, other things being equal, dissuading states from participating in atrocity prevention and reaction through an increase in costs and risks can end up endangering lives by increasing the occurrence of atrocity crimes.

### 2.2.3. Increasing Harm

Proportionality is considered an essential principle of international law and the just war tradition, and an important element of both *jus ad bellum* and *jus in bello*. The proportionality rule prohibits attacks "which may be expected to cause incidental loss of civilian life, injury to civilians, damage to civilian objects, or a combination thereof, which would be excessive in relation to the concrete and direct military advantage anticipated." as described in Article 51(5)(b) of Additional Protocol I to the Geneva Conventions of 1949 (Protocol I 1977). According to the *jus in bello* version of proportionality, the expected cost arising from the pursuit of a military target should be weighed against the expected loss of civilian lives and damage to objects and infrastructure and must be abandoned should the latter outweigh the former. To put it simply, "proportionality requires that the bad effects of such an act not be excessive in relation to the good" (McMahan 2017, p. 131) and that we need to use "force appropriate to the target" (Orend 2013, p. 125). Walzer, talking about proportionality in war, notes that "the rules of war rule out [ . . . ] purposeless or wanton violence" (Walzer 1977, p. 129). Thus, respecting the *in bello* proportionality principle requires a calculation of potential harm, "typically in the form of collateral damage estimates (CDEs)" (Haque 2017, p. 207). While the violation of the *in bello* proportionality principle in the form of excessive collateral damage has been examined extensively, what has received less attention is the risk of violation of *in bello* proportionality through increasing harm as a result, on the one hand, of an urgency to achieve a military victory and, on the other hand, an obligation to conduct and an overconfidence in the success of a post-conflict reconstruction.

The conditions under which belligerents can pursue military victories have been examined in detail, even though there is not always consensus in the various parameters. What the overwhelming majority of scholars agree with though, is that there are constraints on the ways military victory can be achieved. As Walzer argues, "[b]elligerent armies are entitled to try to win their wars, but they are not entitled to do anything that is or seems to them necessary to win. They are subject to a set of restrictions that rest in part on the agreements of states but that also have an independent foundation in moral principle" (Walzer 1977, p. 131). According to the conventional and morally acceptable interpretation, belligerents seek military victory while trying to minimize the cost both to their forces and civilian population (Haque 2017, p. 200) and in the case of humanitarian interventions, "without undermining humanitarian outcomes" (Bellamy 2006, p. 213). What remains to be examined in conjunction with the pursuit of military victory is the risk of increasing harm should there be an obligation to conduct, and an overconfidence in the success of, a post-conflict reconstruction. It can be argued that, by overstating the potential for successful rebuilding and not focusing on the "first do no harm" principle, there is an increased likelihood for a violation of *in bello* proportionality by inflicting disproportionate harm.

Proponents of the responsibility to rebuild are often accused of hubris, of ignoring the potential challenges of rebuilding, and of overestimating the ability of states to suc-

cessfully reconstruct. As Paul Robinson argues, we often operate under the illusion of an ideal situation "in which we know how to do such rebuilding and are able to carry it out successfully" (Robinson 2013, p. 106). The combination of the duty to rebuild and the overconfidence in our ability to successfully do so can lead to a violation of *in bello* proportionality by inadvertently nudging interveners to use disproportionately forceful measures to achieve their military goals in the belief that *post bellum* reconstruction will rectify any injustice incurred. The existence of the responsibility to rebuild is also often justified by the degree of destruction in the target country. As Welsh and Gheciu note "military force has damaging and destabilizing effects on local people and institutions", so "there is a moral injunction on those choosing to engage in military action to stay and ensure that economic devastation and personal insecurity are ameliorated" (Gheciu and Welsh 2009, p. 124), with Walzer underlining that "the work of the virtuous is never finished" (Walzer 2004, p. 21). However, the level of destruction can also be a sign of violation of *in bello* proportionality and of disregarding the rules of *jus in bello.* As Robinson underlines, "if the destruction was so great as to place a country in a position of anarchy where it requires rebuilding, we must doubt whether the war abided by the rules *of jus in bello*, since it seems more than likely that the force used was disproportionate" (Robinson 2013, p. 111). There is clear evidence that often the force used is disproportionate. In Iraq, a commander has noted that the disproportionate use of force by the Americans was somewhat intentional. "[W]e're going to prosecute the war not holding one hand behind our back. When we identify positively an enemy target, we're going to go ahead and take it out with every means we have available. I like to remember what Viscount Slim said during the Burma campaign. He said, 'Use a sledgehammer to crush a walnut.' And that's exactly what we will do." (Brahimi 2010, p. 79). The disproportionate use of force in Iraq was attested by a senior British Army officer, who noted that "the Americans' use of violence is not proportionate and is over-responsive to the threat they are facing" (Brahimi 2010, p. 80). Moreover, the inclusion of a reconstruction element, may inadvertently "sanitize" war and create the illusion that its destructive nature will be diminished. "[W]ar becomes an opportunity for rebuilding" and reconstruction is linked to "the absolution from blame for destruction" as MacGinty has argued (Mac Ginty 2003, p. 613). Civilian casualties and damage to infrastructure in the case of the Iraq war were, on the one hand, compared with the damage that the government of Iraq would do to its population and, on the other hand, downplayed by the anticipated positive effects of post-conflict reconstruction. It thus becomes apparent not only that the need for rebuilding may be an indication for a violation of *in bello* proportionality, but also that the addition of an obligation to conduct and an overconfidence in the success of a post-conflict reconstruction may constitute a slippery slope towards a violation of the rules of *jus in bello*.

*2.3. Legitimacy, Self-Determination, and Human Rights Protection*

There is a general consensus on the fact that in cases of atrocity crimes, that is genocide, war crimes, crimes against humanity, and ethnic cleansing, it is not only permissible that other states intervene to halt this gross violation of basic human rights, but that it constitutes a moral duty to do so. This can be justified on the grounds that a state is legitimate only to the extent that its members enjoy justice-related benefits (Buchanan 2003, p. 247). When this principle is violated, self-determination is outweighed. However, what remains to be explored is the relationship between legitimacy and self-determination *post bellum*. The main question that needs to be answered is the following: is self-determination in post-conflict reconstruction essential for humanitarian interventions to be considered legitimate and for them to be effective? In this section it will be shown that the aforementioned question should be answered affirmatively, and it will be argued that although the violation of self-determination is permissible *in bello* in the case of humanitarian interventions to halt atrocity crimes, disregarding it *post bellum* is not only morally indefeasible but practically ineffective, as it can lead to the delegitimization of the operation and the undermining of human rights protection.

The concept of legitimacy is notoriously complex and can be defined in various ways. For the purposes of post-conflict reconstruction, legitimacy can be said to concern not only the content of *post bellum* policies to be followed but also the procedure and who is entitled to decide what rules and policies should be followed. Thus, legitimacy requires a definition that combines elements of justice (concerning the content of the policies to be followed) with elements of self-determination (concerning the procedure and the actors deciding the policies). Stilz's definition of legitimacy (Stilz 2019, p. 90) is the most accurate and appropriate concerning legitimization processes in post-conflict reconstruction, as she argues that "legitimacy has two dimensions. To be legitimate, a state must not only provide minimum conditions of justice to its population (basic justice); it must also satisfy their interest in being authors of their political institutions (collective self-determination)". What is crucial and highly relevant for the purposes of post-conflict reconstruction is the argument that the second dimension, the "maker" dimension, concerns people's interest "in seeing themselves as the authors of their political institutions" (Stilz 2019, p. 93). The provision of minimally just political institutions, lacking the "maker" dimension of legitimacy, may thus not be a sufficient condition for a state to be considered a legitimate sovereign, as some scholars have suggested (Buchanan 2003, p. 236). What this suggests is that a *post bellum* lack of self-determination can delegitimize post-conflict reconstruction, as people will not see themselves as authors of their political institutions. The responsibility to rebuild comprises thus a fatal flaw: by placing the emphasis more on the content (the conditions that need to be reconstituted by international agents) and less on the procedure (authority will be transferred to local authorities), the current interpretation of the responsibility to rebuild renders humanitarian interventions prone to delegitimization.

Scholars arguing against the prioritization of self-determination contend that a human-rights violating regime, which is not minimally just, has no right to govern and should thus be overthrown, adding that self-determination is not an end in itself and that forcible post-war regime change can be considered a prerequisite for a just post-war state of affairs (Orend 2013, p. 225). This argument, on the one hand goes against the ICISS report, which notes that "overthrow of regimes is not, as such, a legitimate objective" and instead the goal should be to disable the "regime's capacity to harm its own people" (ICISS 2001, p. 35). On the other hand, this argument fails firstly to distinguish between *in bello* and *post bellum* violation of self-determination and secondly to consider the right of individuals, especially those not supporting the regime or being involved in human rights violations, to manage *post bellum* their own affairs without unjustified violent or coercive intervention by others.

The distinction between *in bello* and *post bellum* violation of self-determination is essential and has largely been ignored in the literature. It is correctly argued that a human-rights violating regime engaging in atrocity crimes forfeits its right to self-determination, as the rights at stake far outweigh the value derived from the right of self-determination. Thus, *in bello* violation of self-determination is justified. Once these rights have been protected though, even temporarily, through the cessation of atrocity crimes, *post bellum* violation of self-determination cannot be justified for moral and practical reasons, with these reasons being overlapping and mutually reinforcing. The moral argument concerns, as was briefly mentioned above, the right of individuals to manage *post bellum* their own affairs without unjustified violent or coercive intervention by others. As David Rodin has suggested "in a situation of humanitarian intervention [ . . . ] members of a community who are not engaged in violent or otherwise unjust political action retain the sovereign right to manage their own affairs without interference" (Rodin 2014, p. 250). Stripping the members of a community of the right to manage *post bellum* their own affairs without interference would amount to a violation of rights, additional to the one they have likely been subjected to by their own regime. The *in bello* violation of self-determination is only justified to the extent that it is necessary to ensure the cessation of atrocity crimes, with intervention, per Rodin and contrary to Mill and Walzer, functioning as a way to "protect a space *from* violent politics" (Rodin 2014, p. 257). A *post bellum* violation of self-determination is not only futile, but also morally questionable. The practical implications of a *post bellum* violation of

self-determination concern delegitimization and accusations of imperialism. Regarding the latter, it needs to be noted that overly emphasizing content as opposed to procedure may open interveners to criticism of imperialism and neocolonialism, as according to critics these arguments approximate those of governments that repress their citizens, which often claim that doing so will advance the general welfare, as Pape has noted (Pape 2012, p. 50). Given that the principle of rights vindication precludes the use of coercion after the relevant rights have in fact been vindicated (Orend 2013, p. 189) renders *post bellum* violation of self-determination unjustified and counterproductive, as it "might come to be regarded as a form of 'occupation'" and end up creating "local resistance" (Paris 2014, pp. 576–77) as Roland Paris has argued. Finally, concerning delegitimization, self-determination can be considered a constitutive element of post-conflict reconstruction, not a feature that can be suspended without costs incurred. Thus, rebuilding efforts can be prone to delegitimization irrespective of their success in the case of *post bellum* violation of self-determination, since as Keranen argues (Keranen 2016, p. 12) "imposition of statebuilding reforms is unlikely to result in legitimate and sustainable peace".

Somaliland is a useful case study regarding *post bellum* self-determination and post-conflict reconstruction. In what is arguably the most successful instance of state building in Africa, the international community was not involved at all in rebuilding the state (Englebert and Tull 2008). Despite—or because of—the absence of external statebuilding operations, strong and effective state institutions emerged, with local political and business leaders being in charge of the statebuilding process and post-conflict reconstruction. The locally organized and financed peace conferences managed not only to pave the path for reconstruction, but more importantly to render the people authors of their own political institutions, legitimize the post-conflict reconstruction process, and make Somaliland a de facto state that "enjoys a high degree of legitimacy among its citizens" (Pegg and Kolstø 2015, p. 193). Although donors such as the World Bank often assume that "fragile states lack the capacity for autonomous recovery" (Englebert and Tull 2008, p. 134), the evidence points to the opposite, with highly intrusive statebuilding operations not being necessarily more successful than less intrusive ones, and externally led operations not always perceived by local actors as more legitimate or "as the answers to their problems" (ibid).

### 2.4. Norm Contestation, and Inconsistency

2.4.1. Norm Contestation

The norm of the responsibility to protect has enjoyed impressive attention since its appearance and has traversed the norm's life cycles, from norm emergence to norm cascade to internalization, rapidly (Finnemore and Sikkink 1998, pp. 895–96). Six years after the 2005 World Summit, and merely ten years since the publication of the report by the International Commission on Intervention and State Sovereignty, the then UN Secretary General Ban Ki-moon declared: "By now it should be clear to all that the Responsibility to Protect has arrived" (United Nations 2011). James Pattison notes that RtoP's "central tenets are accepted by the vast majority of (if not all) states, and even some of its strongest critics (e.g., Hehir 2019) now accept that it is a norm (even if "hollow")" (Pattison 2021, p. 891). Apart from the unanimous support to the principle, since its adoption there have been significant attempts to operationalize it with the creation of mechanisms within the UN and individual states.

Its impressive reception, however, can detract from the presence and significance of criticism to the principle. Indeed, both developing and developed countries have expressed concerns about RtoP ever since its emergence. Developing countries were uneasy with the possibility of powerful countries abusing the norm and interfering in their internal affairs, while powerful countries, such as China, Russia, and the United States were reluctant to consent to an obligation to engage their armies in cases not vital to their national interests. Although these initial reservations were overcome, through careful wording of individual states' responsibilities and linking them to a specific set of atrocity crimes and the Outcome Document was unanimously adopted by the General Assembly, a significant number of

countries have continued to voice their concerns towards both the implementation of and the actual norm itself. China, often considered to be leading the criticism against RtoP, is engaging in a form of norm contestation aiming to "reshape the norm's meaning" and challenge "more cosmopolitan notions of conditional sovereignty" (Welsh 2019, p. 61).

Earlier, the threat of states being dissuaded from participating in interventions was underlined, due to their high political and economic cost. Such an event, that is states being reluctant to comply with and support the norm, apart from the obvious consequence of endangering lives, could lead to contestation of the norm (Panke and Petersohn 2016; Deitelhoff and Zimmermann 2019) and potentially its undermining. Two processes can be identified: on the one hand, contestation can be based on states' reluctance, neglect, or failure to comply with and enforce the norm (driven partly by reservations about the norm's validity and the impact of the "mixed motives"[6] and "inconsistency"[7] problems (Paris 2014, p. 578)), and on the other hand, states already critical to the norm can use this opportunity to engage in "substantive contestation" of the norm (Welsh 2013, p. 382), aiming either at redirecting its focus or at weakening its influence.

The example of Libya and Syria, and Russia's subsequent respect of the norm, are indicative of the aforementioned processes. The Libya intervention of 2011, through the invocation of the international community's responsibility to protect, was implicitly supported by Russia, which voted in favor of Resolution 1970 condemning the use of lethal force by the government of Gaddafi and later abstained from Resolution 1973 that authorized all necessary measures to protect civilians. This initial positive stance towards (or at the very least, toleration of) the norm quickly changed as Russia accused the West of violating the mandate of Resolution 1973 through arming the rebels and implementing a regime change (Dagi 2020, p. 380). The apparent existence of "mixed motives" led Russia's Minister of Foreign Affairs Sergei Lavrov to note that Russia's concerns about the implementation of the resolution have been confirmed (Larssen 2016, p. 80), as well as to a diminished trust of the norm by both Russia and China, which, decrying the alleged overreaching, seemed "determined never to 'fall for that trick again'" (Cronoque 2012, p. 128).

Research on norm contestation is relatively recent and studies are still inconclusive about what kind of impact contestation can have on norm robustness (Deitelhoff and Zimmermann 2019; McKeown 2009; Wiener 2014). Contestation can be a result of a state's dissatisfaction with a norm (Havercroft 2017, p. 101) and at the same time contestation can under circumstances also lead to the norm's strengthening, localization, and diffusion (Acharya 2013) and can "resolve contradictions between stakeholders necessary to legitimize international norms" (Jacob 2018, p. 394). As Wiener notes (Wiener 2014, p. 1), contestation "involves the range of social practices which discursively express disapproval of norms". Powerful norm challengers are unlikely to successfully undermine institutionalized norms, with the type of contestation and structural factors being better predictors of a norm's weakening (Deitelhoff and Zimmermann 2019). Norm violation is a necessary but not sufficient condition for undermining a norm. Actors may either continue to comply with the violated norm or engage in sanctioning behavior (Panke and Petersohn 2012, p. 723). A norm can be seriously weakened though, when non-compliance cascades are triggered, that is, when actors routinely violate the norm or actively challenge it with the aim of weakening its influence (Sanders 2018). For our purposes, it is important to identify which instances have, if at all, the potential to undermine RtoP as a norm.

As Welsh has noted, RtoP is "particularly susceptible to contestation, given its inherently indeterminate nature" (Welsh 2013, p. 368). Resistance has predictably been directed predominantly against RtoP's third pillar, concerning the international community's responsibility to take timely and decisive action in case a state is unable or unwilling to protect its population against atrocity crimes. There is already evidence that such contestation can result in states' reluctance to enforce the norm or to "frame situations explicitly in RtoP terms, arguably weakening the political utility of its language" (Welsh 2019, p. 68). The effects of this contestation can lead to, apart from reluctance in the norm's implemen-

tation, limits in its diffusion in the short term and even undermining in the long term (Sanders 2018; Panke and Petersohn 2012). RtoP appears vulnerable to perceptions of politicization, which can lead to noncompliance on the one hand and instrumentalization of the norm by powerful countries for the attainment of political goals on the other. In both cases, the addition of broad responsibilities in the form of the responsibility to rebuild could trigger a noncompliance cascade and potentially norm contestation and weakening.

### 2.4.2. Inconsistency

The responsibility to rebuild can be accused of inconsistency with the rest of the responsibility to protect framework, with that inconsistency risking undermining the norm itself. It can be argued that an a priori unqualified commitment to the responsibility to rebuild undermines the rationale of the responsibility to protect framework, which is built on the premise that military intervention should be taken in case of a breakdown or abdication of a state's own capacity and authority in discharging its "responsibility to protect", as it assumes that the state is unwilling or unable to undertake the task of rebuilding (ICISS 2001). The majority of scholarship on the responsibility to rebuild fails to follow the rationale of RtoP and automatically contends that the responsibility to rebuild always needs to follow a humanitarian military intervention, irrespective of any specific circumstances or the target community's willingness and capacity to engage in *post bellum* reconstruction. This inconsistency can also be related to the aforementioned violation of *post bellum* proportionality. According to the 2005 World Summit Outcome Document adopting the principle of the responsibility to protect, the primary responsibility resides with each individual state, with the role of the international community being that of encouraging and helping states to exercise this responsibility. Only when national authorities manifestly fail in their responsibilities is the international community called to take action. A military humanitarian intervention should not be conceived as *carte blanche* for deep and long-term intervention in the target country and, were the responsibility to rebuild to be incorporated in the responsibility to protect framework, it would need to follow the existing rationale of the RtoP framework, and rebuilding can only be thought of as permissible when the target state is judged to be either unwilling or unable to undertake this responsibility.

### 3. Conclusions

This article aimed to examine the relationship between the responsibility to rebuild and post-intervention reconstruction and to determine whether the current interpretation of the responsibility to rebuild is suitable for attaining the goals of post-intervention reconstruction. It has been shown that, despite the significance of post-intervention reconstruction and the need for a post-intervention strategy, the goals of post-conflict reconstruction can be undermined by the wide-ranging nature of post-intervention obligations and the prioritization of internationally designed and externally directed programs. The aforementioned processes can result in dissuading states from participating in atrocity prevention and inadvertently increasing atrocity crimes, delegitimizing military humanitarian interventions, and inflicting disproportionate harm by violating *in bello* proportionality, while undermining the norm. What has become evident through the analysis here is the need to pursue post-intervention reconstruction by respecting self-determination and avoiding the violation of *in bello* and *post bellum* proportionality. A successful framework will need to combine these elements in order for interventions and post-intervention reconstructions to be effective and legitimate.

**Funding:** This research received no external funding.

**Institutional Review Board Statement:** Not applicable.

**Informed Consent Statement:** Not applicable.

**Data Availability Statement:** Not applicable.

**Conflicts of Interest:** The author declares no conflict of interest.

## Notes

[1] This independent commission was established in September 2000 by the Government of Canada with a mandate to reconcile intervention for human protection purposes and sovereignty.

[2] The terms "post-conflict" and "post-intervention" reconstruction are equivalent and will be used interchangeably.

[3] With the exception of (Robinson 2013).

[4] As Larry May has noted, "like the Preamble of the UN Charter, the responsibility to protect has not yet earned the status of a legal norm in international law. It is thus much more like a moral norm" (May 2013, p. 334).

[5] RtoP was designed and "deliberately institutionalized at the 2005 World Summit as a *political*, rather than legal principle", as Jennifer Welsh notes (Welsh 2019, p. 54). The intention behind the creation of the RtoP was not to create additional legal obligations, but to strengthen states' existing legal commitments.

[6] The inevitable combination of altruistic and self-interested motives in military interventions.

[7] The inconsistent international response to mass atrocities, which can create the appearance of "double standards".

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
