# Peer review of "Post-Intervention Reconstruction and the Responsibility to Rebuild"

_socsci, doi:10.3390/socsci11080368_

Round 1

Reviewer 1 Report

This paper argues that the responsibility to rebuild, as part of the responsibility to protect, is potentially inconsistent with the broader goals of the responsibility to protect. Or so I understand; the specific claim being made could be clearer. 

I take it that what is at stake here is how particular interpretations of the responsibility to rebuild (as part of the RtoP) - in particular, the scope of these responsibilities by intervening parties - may in practice de-legitimize interventions clearly targeted at ending atrocity crimes, either by delegitimizing the intervening parties (who may overstay their welcome, or prevent a quick transfer of authority to local governments), decreasing the willingness of parties to intervene (if governments foresee onerous responsibilities to rebuild) or is otherwise inconsistent with the key objectives of the RtoP.

I think the basic arguments made in the paper are reasonable; very broad interpretations of the responsibility to rebuild could be seen as neo-colonial (and hence illegitimate) and may not result in appropriate outcomes. However, the paper should probably adopt clearer framework - the key issue is the adequate interpretation of the responsibility to rebuild, or rather the scope of those responsibilities and the identification of the responsible agents, not so much the idea of a "responsibility to rebuild" as such; and I suspect more attention to the consequentialist aspects of the argument would be useful as well. For example, the idea that too broad a responsibility to rebuild is self-delegitimizing and hence likely to lead to *less* rebuilding could be seen as a consequentialist argument. 

Author Response

The argument is indeed that particular interpretations of the responsibility to rebuild are undermining its efficacy, and it is not the idea of the responsibility to rebuild that is 'problematic'. The paper argues that the wide-ranging nature of post-intervention obligations and the prioritization of internationally designed and externally directed programs are constitutive elements and priorities of the current interpretation of the responsibility to rebuild, and these undermine the goals of post-intervention reconstruction. 

I hope the argument has been made more clear through the revisions made.

Reviewer 2 Report

The submitted paper presents an interesting summary of parts of the literature on the "R2P"(with particular focus on the responsibility to rebuild) and brings forward some valuable aspects, including the critical aspects of costs (which might have an adverse effect) and aspects of self-determination. Still some critical aspects remain.

1.) while the paper is an interesting literature summary, a question arises about the added value of the paper. In particular the argument, deducted from literature analysis, that the responsibility to rebuild can have adverse effects and actually increase atrocities would have merited an engagement with actual cases. Do we actually see the effects you theoretically anticipate in practice? Detailed case studies would be beyond the scope of the paper, nevertheless, it would merit to underline your argument with empirical material and thus go beyond already existing literature.

2) R2P and thus also R2R are emerging norm in international law. The paper talks about a "norm", but completely leaves aside aspects of international law, where the principle is heavily discussed and this is important, because some arguments in the paper miss the point without taking into account the legal dimension of the norm. First, the "responsibility" to protect is not an obligation to protect, it is not a "legal" norm in international law, it is an emerging norm and a political principle. Important aspects to discuss would be how is it applied, which core obligations do we see emerging, who has the responsibility to rebuild: the intervener, the aggressor, the international community?What actually does rebuilding mean? Is is about "effective rebuilding", meaning e.g. the creation of functioning institutions, or is it sufficient to try, even if the institutions do not sufficiently work in practice? Also when it comes to norm contestation international law is relevant? For customary international law to be established (and there is not yet an "institutionalised" norm of R2P, not from the perspective of international law) we need the consistent and repeated conduct of states with the belief that they act according to the law. The International Court of Justice in its statutes defines it as "evidence of a general practice accepted as law". The inconsistency and the contestation clearly speaks to the fact that we are still far from having a norm of international law. And here again the actual practice and how it is interpreted by states engaging in these activities becomes relevant.

3) The necessity for self-determination in the rebuilding process is an important argument put forward and again the summary of the literature on this issue is highly interesting to read. But who than carries the responsibility to rebuild? How can self-determination be allowed without putting also the obligation to rebuild on a local population, which probably does not have the capacities? One claim made by critical scholars sometimes also is that "local ownership" is a typical neo-liberal argument, putting the responsibility for sufficient development on local societies, while at the same time setting the standards (and direction) for this development from the outside and blaming the locals as deficient when not achieving the standards. Again, a closer look at the effects in practice could add some flesh to the theoretical literature review.

Author Response

  1. Adding empirical material was indeed a weakness in the original version of the article, and in the updated version, I have tried to add as much of it as possible, to show evidence from previous instances of postconflict reconstruction. The main reason why this was not the case in the original article is that the responsibility to rebuild, as a framework, has not been 'implemented', has not been part of an R2P operation thus far,  so we can only use its proclaimed priorities and evidence from past instances of postconflict reconstruction to approximate the effect that the responsibility to rebuild would have, were it to be implemented. Having said that, I think that adding the empirical evidence does indeed help the article clarify its main points.
  2. The R2P is a political, rather than a legal principle, and the main goal behind the creation of the R2P was not to create additional legal obligations for states, but to remind states of their existing obligations, as it has been argued (and as I hope to have clarified through the revisions made). Exploring the international legal dimensions of the R2P is beyond the scope of this article, as the main aim of the article is the examination of the practical implications of the responsibility to rebuild on postconflict reconstruction, the effect it would have on states' behaviour and the likelihood of its success. R2P can thus be considered a norm to the extent  that it changes - or has changed - expectations of states' behaviour when faced with instances of atrocity crimes. 
  3. This is a very good point, and something that I have been thinking about myself. Indeed, the responsibility to rebuild can exceed the capabilities of a local population, especially when these expectations are wide-ranging and externally determined. That is exactly what this paper argues, that for the responsibility to rebuild to be effective, local populations need to be given both the space and the means to rebuild. What the paper tries to show is that post bellum selfdetermination should not be an afterthought, or considered a 'luxury', but is an essential element for the success of postconflict reconstruction, the 'procedure' is as important, if not more so, as the 'content'. I hope the addition of an empirical case has helped illustrate that.

Round 2

Reviewer 2 Report

After the revisions made and in particular the inclusion of some empirical material and case studies, the article provides an intersting debate to the mostly legal debate from a political science perspective. I still think that an engagement with the legal background is an important aspect, but the framing is now clearer and also more in line with current legal interpretations.